# Explicit Thermal Resistance Model of Self-Heating Effects of AlGaN/GaN HEMTs with Linear and Non-Linear Thermal Conductivity

**DOI:** 10.3390/ma15238415

**Published:** 2022-11-25

**Authors:** Surajit Chakraborty, Walid Amir, Ju-Won Shin, Ki-Yong Shin, Chu-Young Cho, Jae-Moo Kim, Takuya Hoshi, Takuya Tsutsumi, Hiroki Sugiyama, Hideaki Matsuzaki, Hyuk-Min Kwon, Dae-Hyun Kim, Tae-Woo Kim

**Affiliations:** 1Department of Electrical, Electronic and Computer Engineering, University of Ulsan, Ulsan 44610, Republic of Korea; 2Korea Advance Nano Fab Center, Suwon-si 16229, Gyeonggi-do, Republic of Korea; 3NTT Device Technology Laboratories, NTT Corporation, Kanagawa 243-0198, Japan; 4Department of Semiconductor Processing Equipment, Semiconductor Convergence Campus of Korea Polytechnics, Anseong-si 17550, Republic of Korea; 5School of Electronics Engineering, Kyungpook National University, 80 Daehak-ro, Buk-gu, Daegu 41566, Republic of Korea

**Keywords:** AlGaN/GaN, self-heating phenomenon, modeling, substrates, thermal resistance

## Abstract

We presented an explicit empirical model of the thermal resistance of AlGaN/GaN high-electron-mobility transistors on three distinct substrates, including sapphire, SiC, and Si. This model considered both a linear and non-linear thermal resistance model of AlGaN/GaN HEMT, the thickness of the host substrate layers, and the gate length and width. The non-linear nature of channel temperature—visible at the high-power dissipation stage—along with linear dependency, was constructed within a single equation. Comparisons with the channel temperature measurement procedure (DC) and charge-control-based device modeling were performed to verify the model’s validity, and the results were in favorable agreement with the observed model data, with only a 1.5% error rate compared to the measurement data. An agile expression for the channel temperature is also important for designing power devices and monolithic microwave integrated circuits. The suggested approach provides several techniques for investigation that could otherwise be impractical or unattainable when utilizing time-consuming numerical simulations.

## 1. Introduction

Owing to their high frequency and power handling potentialities, AlGaN/GaN high-electron-mobility transistors (HEMTs) are expected to play substantial roles in future satellite and information technologies [1,2,3,4]. The majority of the power of such devices is dissipated over relatively small areas of about 0.5–1 μm around the gate contact, resulting in local Joule self-heating [5,6,7,8,9,10]. The performance of a device is usually influenced by self-heating; this can be identified by evaluating the thermal impedance on various epi-structures and substrates (Si, SiC, and sapphire) [11,12,13]. The sapphire substrate, when compared to SiC and Si, exhibits exceptional self-heating effects, with an increase in gate voltage [14,15,16,17]. On the other hand, excessive power density increases the risk of high-power dissipation and high operation channel temperature, both of which have a detrimental effect on the performance and reliability of GaN HEMTs [18,19,20,21,22,23]. Consequently, it is critical to determine the thermal effects. There are a few reports in the literature concerning research into thermal resistance [24,25,26]. Numerous complex models have been introduced, some of which are based on physics and others which are empirical [27,28,29,30,31,32].

Darwish et al. [13] proposed a thermal resistance calculation method for multiple gate fingers. For single-gate HEMTs, Masana [33,34] proposed a gate-angle-related thermal resistance calculation that requires a huge number of estimates, many different components, and a complex model with various parameters. As a result, a concise thermal model for HEMTs is necessary for efficient computation and initial investigation. In order to anticipate values that are close to the findings of the measurements, this study illustrates one such simplified thermal resistance model, with a Taylor series expansion for the power dissipation function. To validate the modeled data, another thermal resistance charge-control-based model was applied. Comparing the thermal resistance values between the DC channel resistance measurements, the extracted thermal resistances from the charge-control model, and our proposed method provided adequate findings. To the best of our knowledge, this is a pioneering work on a simple and reliable empirical model for primary thermal resistance calculations of HEMTs considering both constant and non-linear thermal conductivity.

## 2. Technology and Thermal Measurements

The AlGaN/GaN HEMT structures used in this research were manufactured on 430 μm sapphire, 389 μm 4H-SiC, and 625 μm Si wafers, each 3 inches in size, using the MOCVD technique. The cross-sectional diagram is shown in Figure 1. The epi-structures consist of an 8 nm Al_0.45_Ga_0.55_N barrier layer, a 420 nm channel layer, and a 270 nm GaN buffer in the SiC; a 28 nm Al_0.21_Ga_0.79_N barrier layer, 50 nm channel layer, and 200 nm AlGaN buffer in the Si; a 28 nm Al_0.25_Ga_0.75_N barrier layer, a 150 nm channel layer, an AlN nucleation layer (the thickness is very thin and is not shown in cross sectional diagram), a 200 nm GaN buffer, and a 2.6 µm high-resistance GaN layer in the sapphire. The Schottky contact was formed using Ni/Au, while the ohmic contacts for the source and drain were created using Ti/Al/Ni/Au by e-beam evaporation, followed by annealing at 900 °C for 1 min in a nitrogen environment. This process was the same for all samples. With the support of a Keysight 1500 semiconductor parameter analyzer, the I–V characteristics were measured. Thermal analyses were then conducted using a Temptronic TP03000 thermo-chuck controller.

### Thermal Resistance Model

Three HEMT structures, on different substrates, are shown in Figure 1a–c, with a highly localized heat source under the gate, as shown in Figure 1d, and we assume the device area (length *L_g_* × width *W_g_*). In each case, the AlGaN barrier layer thickness is insignificant and is not anticipated to be a factor in the additional thermal resistance. The thermal conductivities of the majority of semiconductor materials, such as Si, GaAs, and GaN, decrease with increasing temperature [35,36,37,38]. As a necessary consequence, the effects of temperature-dependent thermal conductivity contribute an additional temperature increment that should be considered in the thermal analysis of GaN-based electronics. The nonlinear heat conduction equation for the temperature-dependent thermal conductivity can be solved using finite element analysis (FEA) models [39,40,41,42]. However, the computation times are far greater than those for the linear problem with constant thermal conductivity. In order to address the complications arising with steady-state conduction heat transfer, Kirchhoff’s thermal conductivity is temperature-dependent and is introduced as function *U* as the basis for an integral transform [43]:(1)$$U=K\{T\}=\int_{}^{T}k(\tau )d\tau$$

The findings by Joyce [44] explicitly stated that the evident temperature can be expressed as
(2)$$\theta =T_{0}+\frac{1}{k_{0}}\int_{T_{0}}^{T}k(\tau )d\tau$$
where *T*_0_ is the boundary temperature of the heat sink in the context of the electronic thermal spread complications. If the temperature difference between the channel and substrate (bottom) of the chip is ∆*T*, then Kirchhoff’s transform can be rewritten as
(3)$$\Delta T=\frac{1}{k_{0}(T)}\int_{T_{0}}^{T}k(T^{’})dT^{’}$$
where *k*(*T*_0_) is the thermal conductivity at the backside contact temperature *T*_0_. Hence, a closed-form expression for the channel temperature can be determined using Kirchhoff’s transformation, as noted by Canfield et al. [44,45]:(4)$$\frac{\Delta T}{T_{0}}=\frac{1-{\left(1-\frac{P_{diss}}{4P_{0}}\right)}^{4}}{{\left(1-\frac{P_{diss}}{4P_{0}}\right)}^{4}}$$
where $P_{0}$ is denoted by
(5)$$P_{0}=\frac{\pi k(T_{0})W_{g}T_{0}}{\ln(\frac{8t_{sub}}{\pi L_{g}})}$$
where *P_diss_* is the power dissipation, *L_g_* is the gate length, *W_g_* is the gate width, and *t_sub_* is the substrate thickness. To obtain a clearer approach, the preceding equation can be illustrated as [46]:(6)$$T_{ch}=\left[\frac{1-{\left(1-\frac{P_{diss}}{4P_{0}}\right)}^{4}}{{\left(1-\frac{P_{diss}}{4P_{0}}\right)}^{4}}T_{sub}\right]+T_{sub}$$
and
(7)$$P_{0}=\frac{\pi k(T_{sub})W_{g}T_{sub}}{\ln(\frac{8t_{sub}}{\pi L_{g}})}$$

For AlGaN/GaN HEMTs, this modeling equation estimates the channel temperature *T_ch_* within a scale of feasible values. In our case, we have used AlGaN/GaN HEMTs grown on three different substrates, namely sapphire, Si, and 4H-SiC wafers, for determining the channel temperature. Next, Equation (7) (above), is modified into temperature dependence thermal conductivity using Kirchhoff’s transformation, depicted as [25]
(8)$$k(T)=k_{T_{0}}{\left(\frac{T}{T_{0}}\right)}^{-\alpha }$$
where *α* is the constant, and *k_T_*_0_ is the thermal conductivity at temperature *T*_0_. The value of *α* is one for perfect crystal [25]. Putting this value of *k*(*T*) into Equation (7), *P*_0_ can be written as
(9)$$P_{0}=\frac{\pi k_{T_{0}}W_{g}{\left(T_{sub}\right)}^{1-\alpha }T_{0}{}^{\alpha }}{\ln(\frac{8t_{sub}}{\pi L_{g}})}$$

Although the channel temperature and dissipation power determine the thermal resistance, accurate channel temperature determination is necessary in order to precisely estimate the thermal resistance. First, we performed the DC channel temperature measurement technique noted in [47] and compare the measured results with the modeling Equation (6) for all three substrates.

## 3. Experimental Results and Discussion

Figure 2a–c depict the typical I–V characteristics (output) of the sapphire, SiC, and Si substrates based HEMT, respectively at room temperature. It is clearly observed that the sapphire substrate shows a more negative differential resistance than either Si or SiC at the saturation region with an increase in gate voltage (*V_GS_*) because of the device’s self-heating effects. Self-heating occurs when the added power to the device generates heat that is not efficiently conducted away, thereby allowing the device to remain at the substrate’s ambient temperature [48]. When the drain bias is high, self-heating effects enhance the device’s lattice temperature and degrade physical properties, including mobility (*μ* (m^2^/V ∙ s)) and carrier saturation velocity (*V_SAT_*) [49,50,51,52]. The mobility decreases with increasing temperature as ${\left(1/T\right)}^{2.3}$, with a resulting decrease in DC and RF performance [53]. Although we are interested in heat dissipation, we plotted the drain current (*I_ds_*) as a function of the power (*W*/mm) applied to the device, rather than the bias. The saturated drain current (*I_dsat_*) at each gate bias is then measured; the present curves are then normalized and redrawn as a function of the added power, as shown in Figure 2d–f. The normalization value of the drain current (*I_ds_*) is selected from the maximum saturated drain current (*I_dsat_*). The red dashed line indicates the self-heating boundary limit. For various gate voltages (*V_GS_*), the self-heating incident is clearly observable. In the case of sapphire, self-heating is obvious from *V_GS_* = 0 V to 2 V, and no self-heating is detected at *V_GS_* = −1 V, which is outside the red line (Figure 2a). Consequently, SiC shows self-heating effects at *V_GS_* = 2 V (Figure 2b), and Si indicates self-heating at *V_GS_* = 1 V to 2 V (Figure 2c). In order to determine the channel temperature, we analyzed the temperature dependence of the drain current [54,55], as depicted in Figure 3.

For approximation of the channel temperature without measurement, we evaluated the model using Equation (6). All practical parameters are used in the modeling. For simplicity, we showed the modeling for only the sapphire substrate in Figure 4. There is a large discrepancy between the previously modeled data (shown in red) and the measured data (black circle). As modeling parameters, the following values are used: substrate thickness, *t_sub_* = 430 μm; thermal conductivity of the substrate, *k_sub_* = 49(27/*T_sub_*) W/m-C; gate length, *L_g_* = 14 μm, *T*_0_ = 25 °; and gate width, *W_g_* = 50 μm.

The thermal conductivity of GaN is negligible because its thickness is lower, as compared to the substrate thickness. In the modeled (previous) data, the junction temperature *T_ch_* is overestimated, showing a large discrepancy with the practical results. Considering this, we developed a novel modeling approach that is empirical in nature, but which can be substantiated in terms of the thermal modeling assumption. Here, we review the Equation (6) again:(10)$$\frac{\Delta T}{T_{sub}}=\frac{1-{\left(1-\frac{P_{diss}}{4P_{0}}\right)}^{4}}{{\left(1-\frac{P_{diss}}{4P_{0}}\right)}^{4}}={\left(1-\frac{P_{diss}}{4P_{0}}\right)}^{-4}-1$$

We can use the Tylor series formula for the expansion of the mathematical term
$${\left(1-\frac{P_{diss}}{4P_{0}}\right)}^{-4}$$

This term can be rewritten as,
(11)$$\begin{array}{l} =4\left(\frac{P_{diss}}{4P_{0}}\right)+10{\left(\frac{P_{diss}}{4P_{0}}\right)}^{2}+20{\left(\frac{P_{diss}}{4P_{0}}\right)}^{3}+35{\left(\frac{P_{diss}}{4P_{0}}\right)}^{4}\\ =\frac{P_{diss}}{P_{0}}+0.63{\left(\frac{P_{diss}}{P_{0}}\right)}^{2}+0.313{\left(\frac{P_{diss}}{P_{0}}\right)}^{3}+0.14{\left(\frac{P_{diss}}{P_{0}}\right)}^{4} \end{array}$$

The first and second terms of this expansion series show a quadratic non-linear fit, and the other terms can be disregarded. We rearrange the thermal model equation in the expression below:(12)$$T_{ch}=T_{sub}\frac{P_{diss}}{P_{0}}+\lambda_{1}{\left(\frac{P_{diss}}{P_{0}}\right)}^{2}+T_{a}$$
where $\lambda_{1}$ is the polynomial coefficient, and *T_a_* is the ambient temperature. The 1st term and the 2nd term will be used for linear and non-linear thermal conductivity, respectively. Figure 5a–c depicts the linear and nonlinear calculation (based on thermal conductivity) of all the samples. First, we calculated $P_{0}$ from Equation (9), with both linear and non-linear thermal conductivity. With constant thermal conductivity, $P_{0}$ is all over constant. After obtaining the channel temperature ($T_{ch})$ linear relationship with the dissipated power, $P_{0}$ is again calculated for non-linear thermal conductivity. Table 1 shows the over-all process of calculation. The thermal conductivity of sapphire, which was used for the calculation, is given below [26]:(13)$$k_{sapphire}(T)=49{\left(\frac{T}{27}\right)}^{-1}W/m-C\;$$

In our empirical modeling, we calculated one non-linear term and added it to the linear channel temperature, without changing any parameters of thermal conductivity. Table 2 shows the estimation and the quadratic fit where only $\lambda_{1}$ needs to be adjusted. Here, we used $\lambda_{1}$ = 0.63 from original Equation (11). The average percentage of error is approximately ≈1.5% compared to the non-linear channel temperature calculation (Table 1) and our modeling, which is shown in Figure 6 and Table 2.

For various gate lengths ($L_{g}$), gate widths ($W_{g}$) and substrate thicknesses ($t_{sub}$), the channel temperature, as well as thermal resistance, changes, as shown in Figure 5d–f. In the case of the Si sample, changes in channel temperature are very negligible, while the substrate thickness increases, as shown in Figure 5d.

## 4. Modeled and Extraction Data Verification

Based on the design of our sample structure, we developed a 2D analytical thermal model (recommended by Wang et al. [56]) to evaluate the validity of our empirical thermal model. The extraction procedure is provided in Figure 7 through a flowchart. The device energy band diagram and the structures of the AlGaN/GaN HEMTs considered in the present work are shown Figure 7. The basic charge control equation for 2DEG along the channel is obtained from Poisson’s and Schrodinger equations [29,57]. The relationship between the 2DEG concentration *n_s_* and the gate voltage *V_GS_* can be expressed as
(14)$$n_{s}=\frac{\varepsilon }{qd}\left(V_{GS}-V_{off}-E_{f}\right)$$
(15)$$E_{0}=\gamma_{0}n_{s}{}^{2/3}\&E_{1}=\gamma_{1}n_{s}{}^{2/3}$$
where *q* is the electron charge, and *d* and *ϵ* are the total thickness and permittivity of the AlGaN layer, respectively. *V_off_* is the threshold voltage, and *E_f_* is the Fermi energy level with respect to the bottom of the conduction band. *E*_0_ and *E*_1_ are the levels of the two lowest sub-bands. *V_off_* is defined as [57]:(16)$$V_{off}=\varphi_{b}-\Delta E_{C}-\frac{qN_{D}d^{2}}{2\varepsilon }-\frac{\sigma_{pz}}{\varepsilon }d$$
where *φ_b_* is the Schottky barrier height, and *d* is the thickness of the AlGaN barrier. *N_D_* is the doping concentration of the AlGaN layer, *σ_pz_* is the polarization induced charge density, and ∆*E_c_* is the conduction band offset at the AlGaN/GaN interface. A polynomial expression can be used to represent *E_f_* as a function of *n_s_* [58]:(17)$$E_{f}=k_{1}+k_{2}n_{s}{}^{1/2}+k_{3}n_{s}$$
(18)$$n_{s}={\left[\frac{-k_{2}+\sqrt{k_{2}{}^{2}+4k_{3}{}^{’}(V_{GS}-V_{off}-k_{1})}}{2k_{3}{}^{’}}\right]}^{2}$$
where *k*_1_, *k*_2_, and *k*_3_ are temperature-dependent parameters and *k’*_3_
*= k*_3_
*+ qd/ϵ*. Considering three consecutive polynomial expressions, the parameters can be expressed as below:(19)$$k_{3}=\frac{\left(\sqrt{n_{s2}}-\sqrt{n_{s3}})(E_{f1}-E_{f2})-(\sqrt{n_{s2}}-\sqrt{n_{s3}})(E_{f2}-E_{f3}\right)}{\left(n_{s1}-n_{s2})(\sqrt{n_{s2}}-\sqrt{n_{s3}})-(n_{s2}-n_{s3})(\sqrt{n_{s1}}-\sqrt{n_{s2}}\right)}$$
(20)$$k_{2}=\frac{\left(n_{s2}-n_{s3})(E_{f1}-E_{f2})-(n_{s1}-n_{s2})(E_{f2}-E_{f3}\right)}{-(n_{s1}-n_{s2})(\sqrt{n_{s2}}-\sqrt{n_{s3}})+(n_{s2}-n_{s3})(\sqrt{n_{s1}}-\sqrt{n_{s2}})}$$
and
(21)$$k_{1}=E_{f1}-(k_{2}\sqrt{n_{s1}}+k_{3}n_{s1})$$
where *E_f_*_1_*, E_f_*_2*,*_
*E_f_*_3*,*_
*n_s_*_1*,*_
*n_s_*_2_, and *n_s_*_3_ are the three regional states of the Fermi energy levels and the positions of the 2DEG concentrations. Figure 7a–c verifies that the 1st sub-band of all the samples in our model are below the Fermi levels. As seen from the figure, the 2nd sub-band (*E*_1_) is significantly larger than *E*_0_. Therefore, the second sub-band’s contribution to *n_s_* can be omitted [19].

### Current-Voltage Characteristics

In the linear region, the model depends on three temperature parameters that can be expressed as [59]:(22)$$I_{D_{LIN}}=\frac{\zeta }{4k_{4}{}^{2}}[\delta (D_{1})-\delta (D_{2})];\zeta =\frac{q\mu_{0}W}{\left(L+\frac{V_{D}}{E_{C}}\right)}$$
where $\delta (D)=k_{2}{}^{2}D+\frac{D^{2}}{2}-\frac{4}{3}k_{2}D^{3/2}$, *μ*_0_ = low field mobility, and *E_C_* = critical electric field. The values of *D*_1_ and *D*_2_ are defined as
(23)$$D_{1}=k_{2}{}^{2}+4k_{4}(V_{G_{0}})$$
(24)$$D_{2}=k_{2}{}^{2}+4k_{4}(V_{G_{0}}-V_{D})$$
(25)$$V_{G_{0}}=V_{G}-V_{off}-k_{1}$$

In the saturation region, the electron velocity is saturated at *V_SAT_* and is defined by:(26)$$I_{D_{SAT}}=qWV_{SAT}{\left[\frac{-k_{2}+\sqrt{k_{2}{}^{2}+4k_{4}(V_{G_{0}}-V_{D_{SAT}})}}{2k_{4}}\right]}^{2}$$

By introducing the self-heating effect, the total drain current expression can be written as [56]
(27)$$I_{D_{SH}}=I_{D_{SAT}}[1-\frac{\eta (I_{D_{SAT}}V_{DS}R_{TH})}{T_{0}I_{D_{SAT}}V_{DS}R_{TH}+T_{0}{}^{2}}]$$

Here, *η* = fitting parameter = 500 K, and *T*_0_ = absolute temperature. Considering these equations, we modeled the transfer and output characteristics of all the samples, as shown in Figure 8. The parameters used in modeling are shown in Table 3. It can be explained that negative differential resistance has no constant values at different levels of gate voltages. AlGaN/GaN on sapphire suffers from a negative output differential resistance that starts from *V_G_* = 0 V. The modeling data does not cover the negative gate voltages because there is no self-heating effect observed at that voltage level in any of the samples. The SiC and Si devices show insignificant self-heating effects, in contrast to sapphire, as seen in Figure 8e,f. Table 4 displays the charge-control model’s extracted thermal resistances, which are then compared to the results of our model and the measured data.

From Table 4, the extracted thermal resistance values from the charge control model of the sapphire substrate and silicon are high compared to those for SiC. The reason behind this high thermal resistance found in silicon is due to the high substrate thickness (625 μm). We used *η* = 500 K for all calculations. The results could be modified by adjusting the temperature-dependent mobility (*α* ≈ 1.6–1.8) [56].

## 5. Conclusions

An accurate empirical model was used to estimate the thermal resistances of AlGaN/GaN HEMTs. Combining experimental results with data from the charge-control model forecasts favorable results for the validation of this model. The heat resistance levels of three distinct substrates were analyzed and contrasted. The measurements and comparisons encompassed more than 30 devices on each substrate. The issue of overestimating the channel temperature presents difficulties for an accurate computation of thermal resistance using a prior model, which is resolved in this work by utilizing a fundamental mathematical model technique. In future research, this proposed empirical model will be implemented in RF MMIC (monolithic microwave integrated circuit) devices to accurately estimate the channel temperature for better prediction reliability.

## Figures and Tables

**Figure 1 materials-15-08415-f001:**
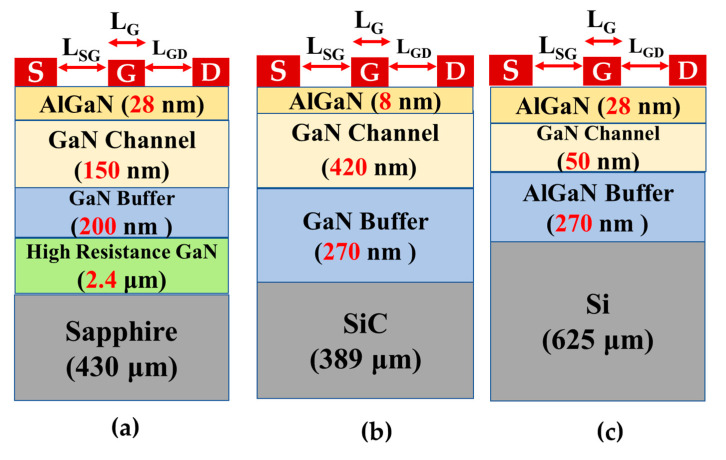
Cross-sectional diagram of AlGaN/GaN HEMT on sapphire (**a**), SiC (**b**), and Si (**c**), with a highly localized heat source under the gate (**d**).

**Figure 2 materials-15-08415-f002:**
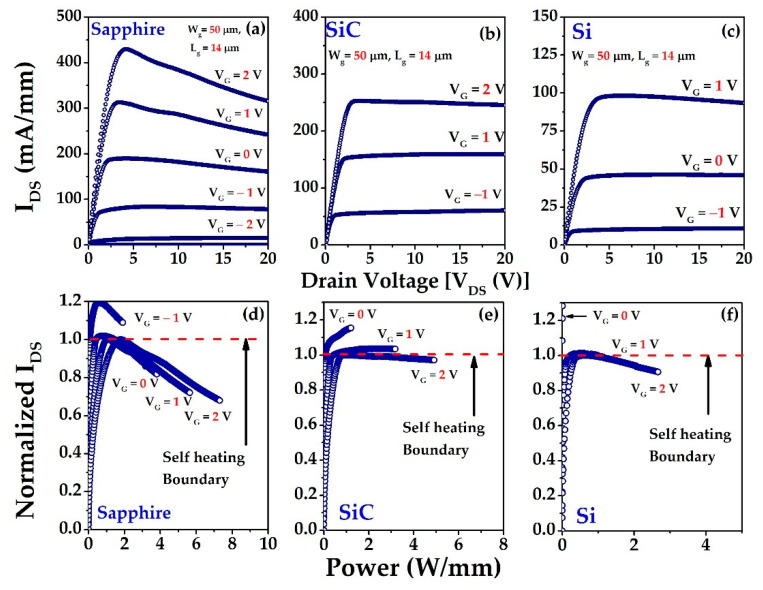
I–V characteristics of (**a**) sapphire, (**b**) SiC, and (**c**) Si at room temperature; the power dissipation and self-heating phenomena of (**d**) sapphire, (**e**) SiC, and (**f**) Si.

**Figure 3 materials-15-08415-f003:**
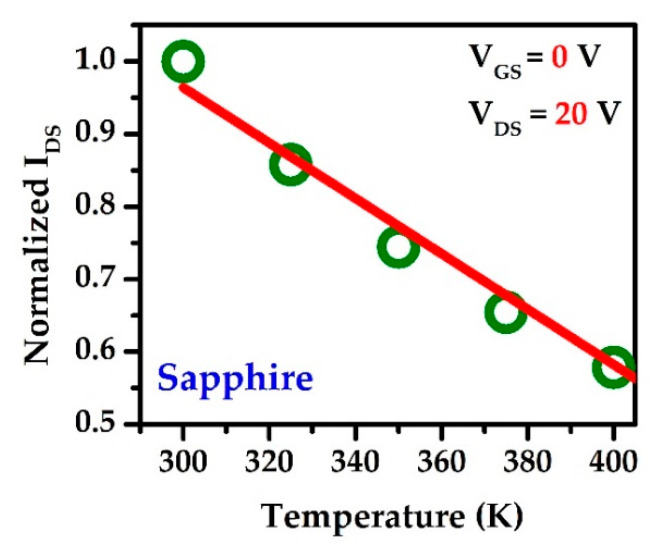
Temperature dependence of drain current at high *V_DS_* value.

**Figure 4 materials-15-08415-f004:**
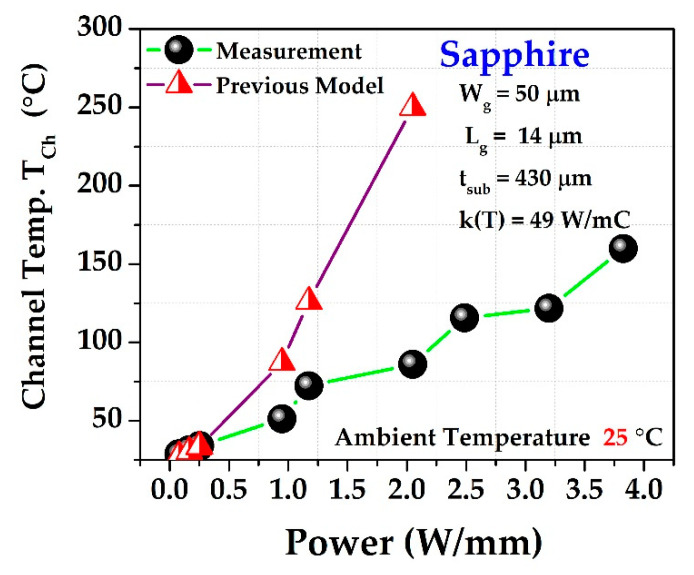
Previous model (marked as half-red triangle) over-estimates $T_{ch}$, which does not match with the experimental results (black sphere).

**Figure 5 materials-15-08415-f005:**
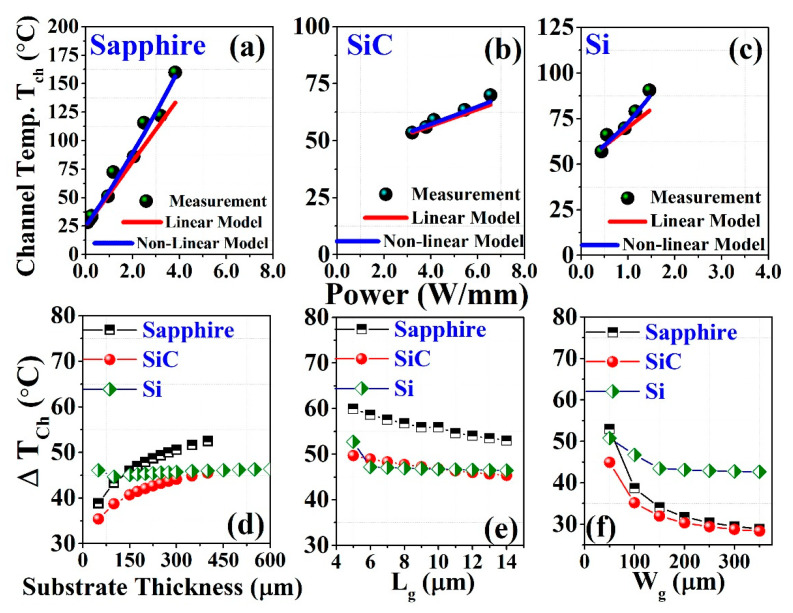
Thermal resistance measurement and the linear and non-linear model of (**a**) sapphire, (**b**) SiC, and (**c**) Si. Thermal resistance dependence of substrate thickness (**d**), gate length (**e**), and gate width (**f**).

**Figure 6 materials-15-08415-f006:**
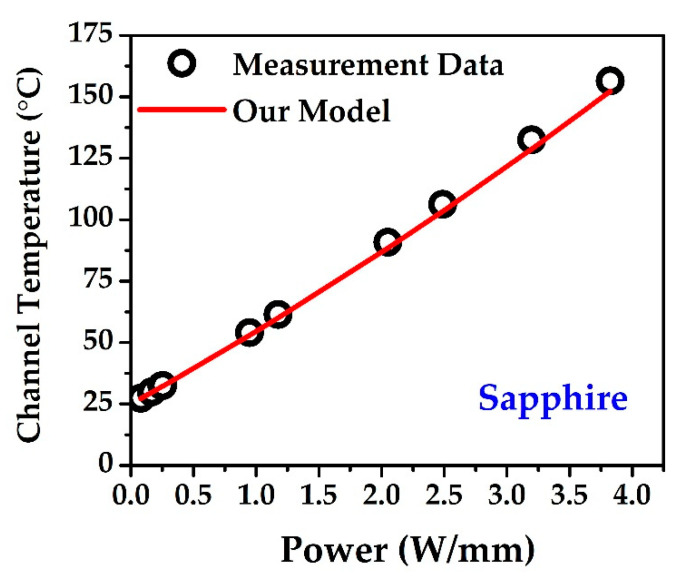
Measurement data (black line and square) and our model (red line and circle). For simplicity, only the sapphire substrate non-linear model is shown here.

**Figure 7 materials-15-08415-f007:**
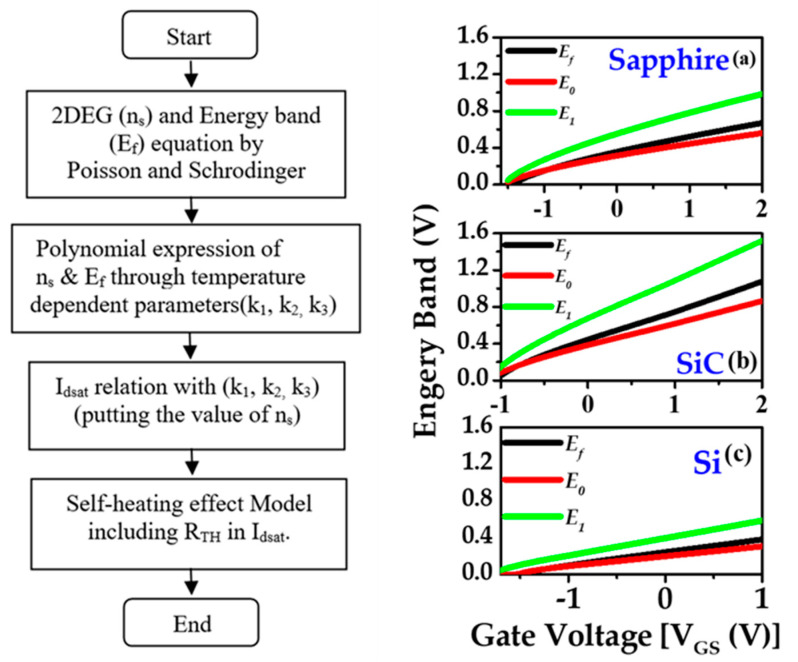
Charge control model flowchart of the self-heating effect and energy bands of sapphire (**a**), SiC (**b**), and Si (**c**).

**Figure 8 materials-15-08415-f008:**
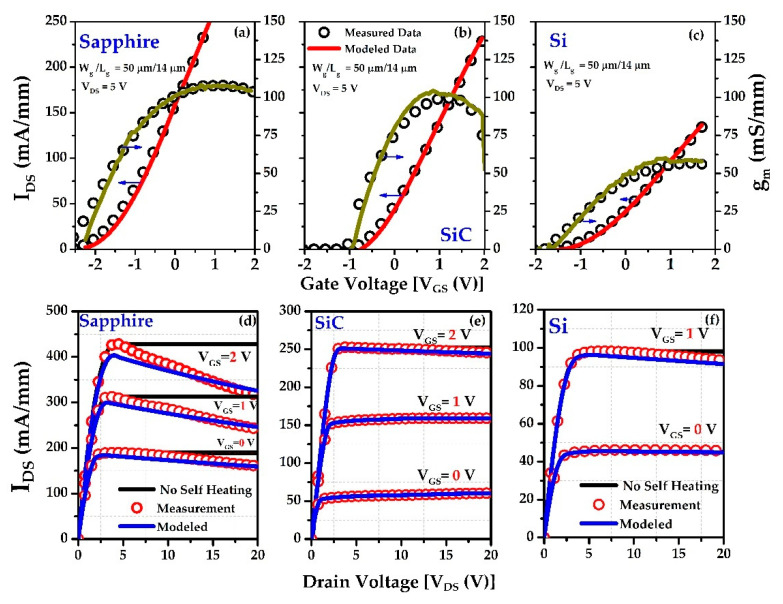
Charge control-based modeling data showing the transfer characteristics of sapphire (**a**), SiC (**b**), and Si (**c**). Modeled output characteristics and measurement data of sapphire (**d**), SiC (**e**), and Si (**f**).

**Table 1 materials-15-08415-t001:** Calculation of linear and non-linear channel temperature (sapphire substrate).

$P_{diss}$	$P_{0}$ [Constant k(T)]	Channel Temperature (T_ch_)	$P_{0}$ [Non-Linear k(T)]	Channel Temperature (T_ch_) [from Non-Linear k(T)]
0.0039	0.0441	27.24	0.0083	27.34
0.0039	0.0441	27.26	0.0095	27.36
0.0082	0.0441	29.63	0.0116	29.86
0.0083	0.0441	29.72	0.0133	29.95
0.0124	0.0441	32.01	0.0189	32.37
0.0127	0.0441	32.18	0.0213	32.55
0.0474	0.0441	51.84	0.0343	54.07
0.0587	0.0441	58.28	0.0345	61.39
0.1024	0.0441	83.09	0.0371	90.79
0.1243	0.0441	95.48	0.0372	106.21
0.1599	0.0441	115.66	0.0405	132.49
0.1913	0.0441	133.42	0.0405	156.46

**Table 2 materials-15-08415-t002:** New model and measurement data (sapphire substrate).

$P_{diss}$	$P_{0}$ [Constant k(T)]	$\frac{P_{diss}}{P_{0}}$	${\left(\frac{P_{diss}}{P_{0}}\right)}^{2}$	Non-Linear Model	Percentage of Error (%) with Measurement
0.0039	0.0441	0.089	0.0080	27.25	0.328
0.0039	0.0441	0.090	0.0082	27.27	0.330
0.0082	0.0441	0.185	0.0343	29.67	0.621
0.0083	0.0441	0.188	0.0356	29.76	0.630
0.0124	0.0441	0.281	0.0787	32.093	0.865
0.0127	0.0441	0.287	0.0825	32.267	0.881
0.0474	0.0441	1.074	1.1530	52.998	1.979
0.0587	0.0441	1.331	1.7724	60.056	2.173
0.1024	0.0441	2.324	5.3994	88.491	2.539
0.1243	0.0441	2.819	7.9499	103.438	2.607
0.1599	0.0441	3.626	13.1513	128.813	2.780
0.1913	0.0441	4.337	18.8081	152.223	2.706

**Table 3 materials-15-08415-t003:** Parameters used in modeling.

Symbol	SiC	Si	Sapphire
*x*	0.45	0.21	0.20
*k* _1_	−0.11	−0.12	−0.11
*k*_2_ (V ∙ cm)	1.76 × 10^−9^	1.96 × 10^−9^	1.74 × 10^−9^
*k*_3_ (V ∙ cm^2^)	1.76 × 10^−18^	1.10 × 10^−18^	1.80 × 10^−18^
*d* (nm)	8	28	28
*V_SAT_* (V/m)	9.5 × 10^8^	9.0 × 10^8^	4.5 × 10^8^
*V_off_* (V)	−0.88	−1.54	−2.63
*μ*_0_ (m^2^/V ∙ s)	0.0126	0.0186	0.016

**Table 4 materials-15-08415-t004:** Comparison of thermal resistance in the DC measurement method, the charge control model, and our proposed model.

Sample	Drain Voltage Range	Measured R_th_ (°C/W)			Charge Control based R_th_ (°C/W) [Average]	Our Model R_th_ (°C/W)
		*V_GS_* = 2 V	*V_GS_* = 1 V	*V_GS_* = 0 V		
Sapphire	10–13 V	645	605	554	630	625
	14–16 V	650	705	729		
	17–20 V	673	714	830		
SiC	13–15 V	136	178	-	150	127
	16–20 V	140	165			
Si	10–13 V	-	593	683	705	712
	14–16 V	-	636	800		
	17–20 V	-	846	905		

## Data Availability

Not applicable.

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
