# Peer review of "Explicit Thermal Resistance Model of Self-Heating Effects of AlGaN/GaN HEMTs with Linear and Non-Linear Thermal Conductivity"

_materials, 2022, doi:10.3390/ma15238415_

Round 1
Reviewer 1 Report
Review Report for Manuscript ID:2053938
Paper is of current interest and falls in the scope of the journal, however, there are the following suggestions authors should address and then I welcome for publication:
1. Abstract should be enhanced with major outcomes.
2. Language needs minor improvements. Please check all manuscript for typo and punctuation mistakes. Grammatical errors and typos that could be seen within the manuscript should be amended.
3. Would you please cite all equations that you did not derive?
4. Discussions should be enriched with physical meaning.
5. The updated literature reference seems to be limited and it should encompass the current level of knowledge.
Following are few suggestions on recent updates
· Nonlinear solar thermal radiation efficiency and energy optimization for magnetized hybrid-Prandtl-Eyring nanoliquid in Aircraft, Arabian Journal for Science and Engineering, 2022.
· A Case study of heat transmission in a Williamson fluid flow through a ciliated porous channel: A semi-numerical approach, Case Studies in Thermal Engineering, 2022.
6. Future works with applicability should be included at the end.
Author Response
Dear Editor,
We are thankful for the constructive comments from the reviewers that we have considered in detail. After discussion and revision, we now submit this revised version of the manuscript to the MDPI Materials. In this letter, we enclose the point-by-point replies to all the valuable comments from the reviewers. We would appreciate your reconsideration of our work.
Point No: 1
Abstract should be enhanced with major outcomes.
Our response:
Dear Reviewer,
Thank you very much for carefully reviewing our manuscript and providing fruitful suggestions. We appreciate the reviewer’s comment. We have made changes in our manuscript according to reviewer’s comment and observation.
Corresponding changes in the Manuscript: Yes, We have added major outcomes in the abstract.
Location of Change: Abstract.
Line:19 to 21, Highlighted in red in the revised manuscript.
Point No: 2
Language needs minor improvements. Please check all manuscript for typo and punctuation mistakes. Grammatical errors and typos that could be seen within the manuscript should be amended.
Our response:
Thank you very much for carefully reviewing our manuscript and notice the typo and errors in the manuscript. We have made changes in our manuscript according to the reviewer’s comments and careful observation.
Corresponding changes in the Manuscript: Yes, We have checked the typo and Grammatical errors.
Location of Change:
Keywords: AlGaN/GaN (line:27), compared to (Line: 37), fig. 1(a),(b),(c), performed (line 144), in-room temperature (line 148), drain current (Ids)[line: 156], power (w/mm)[line:156], saturated drain current (Idsat)[line:161], channel (line: 170), The (line 225), VSAT (line: 288), no (line: 300), does (line: 302), self-heating effect observed on that voltage level in all samples (line 303), this (line: 314), from previous model (line 347). Highlighted in red in the revised manuscript.
Point No: 3
Would you please cite all equations that you did not derive?
Our response:
Thank you very much for carefully reviewing our manuscript and notice the citation of the equation in the manuscript. We have made changes in our manuscript according to the reviewer’s comments.
Corresponding changes in Manuscript: Yes, We added citation of a equation that was not derive by us.
Location of Change:
Section 2: Technology and Thermal Measurement
Sub-section 2.1: Thermal resistance model.
Line: 132.
Section 3: Experimental Results and Discussion
Line: 208.
Section 5: Modeled and Extraction data verification
Line: 239, line: 246.
Point No: 4
Discussions should be enriched with physical meaning.
Our response:
Thank you very much for your comment. We have made changes in our manuscript according to the reviewer’s comments. In the future, we will add more practical examples.
Corresponding changes in the Manuscript: Yes, We have added some information regarding this issue.
Location of Change:
Section 3: Experimental Results and Discussion.
Line: 147, 148, 149, 155-159, 169 ; Highlighted red in the revised manuscript.
Added references : 50-53, 55,56
Point No: 5
The updated literature reference seems to be limited and it should encompass the current level of knowledge.
Our response:
Thank you very much for your comment. We have made changes in our manuscript according to reviewer’s comment.
Corresponding changes in Manuscript: Yes, We have added the mentioned literature to extends our research more consequential.
Location of Change:
Section 1: Introduction
Reference No: 5 and 27.
Line: 34 and 43.
Point No: 6
Future works with applicability should be included at the end.
Our response:
Thank you very much for your comment and fruitful suggestion. We have made changes in our manuscript according to reviewer’s comment.
Corresponding changes in Manuscript: Yes, We have added the future work in the conclusion part.
Location of Change:
Section 6: Conclusion
Line: 348 to 350. Highlighted red in the revised manuscript.
Reviewer 2 Report
Reviewer comments on Manuscript ID 2053938
In the work by Kim and Kim et al, they provided an in-depth analysis of with the use of an explicit thermal resistance model for the self-healing effects of AlGaN/GaN HEMTs with linear and non-linear thermal conductivity which offered several modes of inquiry which was not possible through numerical simulations. The manuscript is detailed and well written, providing the general knowledge and physical equations in supporting the different modelling simulations. However, the following concerns should be addressed so that it could be accepted to be published.
1. In the main text, page 3 line 70 “thickness is very low” sounds strange, perhaps thickness is very thin instead of low to sound more appropriate.
2. In the main text, page 3 line 74 “This process is the same in all samples.” Instead of “This process same in all samples.”
3. In the main text, page 3 line 95, the author quoted a reference from a single author but just stating the name “Joyce [42]” seems strange. Perhaps indicating the findings by Joyce or results reported by them would be more appropriate.
4. In the main text, page 3 line 112, the equation for Po seems to have truncated from the page and inserted below the figure, do ensure that the formatting sequence is proper arranged.
5. There seems to be some formatting and language errors in the main text. In page 6 line 173, “Thermal conductivity of GaN is negligible because of its thickness is low compared to substrate thickness” should be corrected to “Thermal conductivity of GaN is negligible because its thickness is lower as compared to the substrate’s thickness”.
There is also an error in the sequencing of the sections for example, the numbering for section 3 directly skipped to section 5, the sequence of the section should be reviewed to ensure any numbering errors.
Author Response
Dear Editor,
We are thankful for the constructive comments from the reviewers that we have considered in detail. After discussion and revision, we now submit this revised version of the manuscript to the MDPI Materials. In this letter, we enclose the point-by-point replies to all the valuable comments from the reviewers. We would appreciate your reconsideration of our work.
Point No: 1
In the main text, page 3 line 70 “thickness is very low” sounds strange, perhaps thickness is very thin instead of low to sound more appropriate.
Our response:
Dear Reviewer,
Thank you very much for carefully reviewing our manuscript and providing fruitful suggestions. We appreciate the reviewer’s comment. We have made changes in our manuscript according to the reviewer’s comments and observations.
Corresponding changes in the Manuscript: Yes, We have changed the line.
Location of Change:
Section 2: Technology and Thermal Management
Page 3 , Line:71, Highlighted red in the revised manuscript.
Point No: 2
In the main text, page 3 line 74 “This process is the same in all samples.” Instead of “This process same in all samples.”
Our response:
Thank you very much for carefully reviewing our manuscript. We have made changes in our manuscript according to the reviewer’s comments and observations.
Corresponding changes in the Manuscript: Yes, We have changed the line.
Location of Change:
Section 2: Technology and Thermal Management
Page 3 , Line:75, Highlighted in red in the revised manuscript.
Point No: 3
In the main text, page 3 line 95, the author quoted a reference from a single author but just stating the name “Joyce [42]” seems strange. Perhaps indicating the findings by Joyce or results reported by them would be more appropriate.
Our response:
Thank you very much for your comment. We have made changes to our manuscript according to the reviewer’s comments and observations.
Corresponding changes in the Manuscript: Yes, We have changed the sentence.
Location of Change:
Section 2: Technology and Thermal Management
Sub-section: Thermal resistance model.
Page 3 , Line:96, Highlighted red in the revised manuscript.
Point No: 4
In the main text, page 3 line 112, the equation for Po seems to have truncated from the page and inserted below the figure, do ensure that the formatting sequence is proper arranged.
Our response:
Thank you very much for your comment. We have made changes in our manuscript according to the reviewer’s comments and observations.
Corresponding changes in the Manuscript: Yes, We have changed the notation.
Location of Change:
Section 2: Technology and Thermal Management
Sub-section: Thermal resistance model.
Page 3 , Line:112, Highlighted red in the revised manuscript.
Point No: 5
There seems to be some formatting and language errors in the main text. In page 6 line 173, “Thermal conductivity of GaN is negligible because of its thickness is low compared to substrate thickness” should be corrected to “Thermal conductivity of GaN is negligible because its thickness is lower as compared to the substrate’s thickness”.
There is also an error in the sequencing of the sections for example, the numbering for section 3 directly skipped to section 5, the sequence of the section should be reviewed to ensure any numbering errors.
Our response:
Thank you very much for your comment. We have made changes in our manuscript according to the reviewer’s comments and observations.
Corresponding changes in the Manuscript: Yes, We have changed the sentences and corrected the numbering.
Location of Change:
Section 3: Experimental results and discussion
Page 6 , Line:180, Highlighted in red in the revised manuscript.
Location of Change:
Section 5: Modeled and Extraction data verification
Sub-section 5.1: Current-Voltage Characteristics (corrected).
Page 11, Line:276, Highlighted in red in the revised manuscript.
Round 2
Reviewer 1 Report
Recommendations: Accept revision